# Mapping the coevolution, leadership and financing of research on viral vectors, RNAi, CRISPR/Cas9 and other genomic editing technologies

David Fajardo-Ortiz[1]*, Annie Shattuck[2], Stefan Hornbostel[1]

**1** Research System and Science Dynamics research area, Deutsche Zentrum für Hochschul- und Wissenschaftsforschung (DZHW), Berlin, Germany, **2** Department of Geography, Indiana University, Bloomington, Indiana, United States of America

\* fajardo@dzhw.eu, davguifaj@gmail.com

**Data Availability Statement:** All relevant data are within the manuscript and its Supporting Information files.

## Abstract

Genomic editing technologies are developing rapidly, promising significant developments for biomedicine, agriculture and other fields. In the present investigation, we analyzed and compared the process of innovation for six genomic technologies: viral vectors, RNAi, TALENs, meganucleases, ZFNs and CRISPR/Cas including the profile of the main research institutions and their funders, to understand how innovation evolved and what institutions influenced research trajectories. A Web of Science search of papers on viral vectors RNAi, CRISPR/Cas, TALENs, ZFNs and meganucleases was used to build a citation network of 16,746 papers. An analysis of network clustering combined with text mining was performed. For viral vectors, a long-term process of incremental innovation was identified, which was largely publicly funded in the United States and the European Union. The trajectory of RNAi research included clusters related to the study of RNAi as a biological phenomenon and its use in functional genomics, biomedicine and pest control. A British philanthropic organization and a US pharmaceutical company played a key role in the development of basic RNAi research and clinical application respectively, in addition to government and academic institutions. In the case of CRISPR/Cas research, basic science discoveries led to the technical improvements, and these two in turn provided the information required for the development of biomedical, agricultural, livestock and industrial applications. The trajectory of CRISPR/Cas research exhibits a geopolitical division of the investigation efforts between the US, as the main producer and funder of basic research and technical improvements, and Chinese research institutions increasingly leading applied research. Our results reflect a change in the model for financing science, with reduced public financing for basic science and applied research on publicly funded technological developments in the US, and the emergence of China as a scientific superpower, with implications for the development of applications of genomic technologies.

**Funding:** This research was funded through an Alexander von Humboldt Stiftung (http://www.humboldt-foundation.de) postdoctoral grant to David Fajardo-Ortiz (grant number: 1195113). The funders had no role in study design, data collection and analysis, decision to publish, or preparation of the manuscript.

**Competing interests:** The authors have declared that no competing interests exist.

## Introduction

Viral vectors, RNA interference (RNAi), and genomic editing platforms are technologies aimed at understanding and controlling the genomic machinery of biological organisms. Each of these technologies has been presented at the time, in different decades, as revolutionary and promising. Gene therapy, which is mostly based in viral vector technologies as gene delivery systems, has been regarded as a potential revolution in medicine since the 90s.[1, 2] While at least since 2005 RNAi, which was originally discovered as a highly conserved regulatory mechanism of genetic expression, [3] has been considered a revolution in biology "as researchers have sought to understand how RNAi works to regulate gene expression, have used it to perform reverse genetics in mammalian cells, and have begun to explore its potential therapeutic use. [4]" Other authors have pointed out that RNAi "has not only revolutionized our ability to study biology, but has challenged some of the most basic assumptions behind our understanding of it. [5]" Similarly, CRISPR/Cas9, which was originally discovered as part of the innate immune system of many bacteria and archaea [6], has been named as "emerging as a revolutionary technique for genome editing" due to its "economy, high efficiency, precise targeting and flexible technical extension compared with traditional DNA sequence modifying measures [7]," and because CRISPR/Cas9 genome editing can achieved in any species [6].

Besides CRISPR/Cas9, there are other three genome editing platforms: transcription activator-like effector nucleases (TALEN), zinc-finger nucleases (ZFNs) and meganucleases [8]. These three genome editing platforms are considered not just precursors of CRISPR/Cas9 [8] but also incumbent technologies that eventually would be supplanted by CRISPR/Cas9 [9]. Interestingly, viral vectors and RNAi have been also considered incumbent genetic therapeutic technologies with reference to CRISPR/Cas9 [10] which is considered a disruptive innovation [11]. As a matter of fact, the relation between viral vectors, RNAi and CRISPR/Cas9 is more sophisticated. For example, a comparative review of RNAi and CRISPR/Cas9 technologies suggest that even though CRISPR technologies can "make RNAi obsolete," there will be still specific domains of biomedical research and clinical applications for RNAi [12]. On the other hand, the relationship between viral vectors and other technologies is more collaborative than competitive since viral vectors can be used as delivery system for CRISPR [13] and RNAi [14].

Comparing the trajectories of these six technologies (Viral vectors, RNAi and the four genome editing platforms) as well as analyzing the interactions that take place between them is relevant for the general understanding of the current processes of technological invention and development that take place in the biotechnological field. This is due to the different times in which these biotechnologies emerged [1–7], the revolutionary and innovative character attributed to them [1–7] and the competitive and collaborative relationships that exist between these technologies [8–14]. A first step to analyse the coevolution of these technologies is by the identification and analysis of research fronts (clusters of papers in the citation networks). This methodological approach has proved a useful tool to analyse the emergence, structure and evolution of the research on a diversity of technologies like cancer nanotechnologies [15, 16], medical big data [17], regenerative medicine [18] and small-molecule drugs [19]. The maps generated with this methodology integrates the information on the structure, organization and content of the papers published by specialized research communities. However, in order to get a more complete picture of the evolution of the research on these technologies it is important to integrate the metadata from papers referring to the institutions where the investigations were carried out and the information about the agencies that funded these investigations. Incorporating information about research institutions and funding agencies have allowed researchers, for example, to identify the emergence of new geopolitical actors such as South Korea and Turkey in the global networks of scientific research and technological development

[20]. Moreover, this type of analysis has revealed how the different actors that make up national innovation systems such as universities, industries and governments which have different regulations, research goals and incentive structures are articulated on a global scale [21]. More importantly, research funding is an important factor that can influence the structure and content of an entire field of research such as nanotechnology [22].

In the present investigation, we set out to map the trajectories of research on RNAi, viral vectors and genomic editing platforms, as well as identify the interactions between them and determine the profile of the main research institutions and funding agencies. The generated maps allow us to visualize the organization of research on these genomic biotechnologies as a simultaneously scientific and socio-political process.

## Methodology

1. Six separate searches for papers on viral vectors, RNAi and three different genomic editing platforms were performed in the Web of Science [23] during June, 2019. The search criteria for each of the technologies are shown in Table 1:

2. A network model was built with the papers found in the Web of Science by using the software HistCite [24]. Then, the network model was analyzed and visualized with Cytoscape [25].

3. The number of papers per year of each technology was plotted in order to identify their phases of exponential growth, conservation and decay. This information was used to divide the citation network model according to the following time periods: 1966–2000, 2001–2005, 2006–2010, 2011–2015 and 2016–2019. These time periods correspond approximately to the different stages of exponential growth, conservation and decay of research on these technologies. Additionally, this division of the citation network model allowed us to obtain sub-networks of similar size in terms of number of articles.

4. The network models were displayed by using the force-directed layout algorithm of Cytoscape [25].

5. A cluster analysis based in the Newman modularity [26] was performed for every time period using Clust&see, a Cystoscape plug-in [27]. This analysis divided the sub-networks in several clusters or modules of papers. This clusters are defined by Newman as "groups of vertices within which connections are dense but between which they are sparse" [26].

**Table 1. Search criteria for papers on viral vectors, RNAi and different genomic editing platforms in the Web of Science.**

| Technology | Search criteria | Refined by: | Indexes: |
|---|---|---|---|
| CRISPR/Cas9 | Title: (CRISPR or "clustered regularly interspaced short palindromic repeats" or Cas9) | Document type: Article | SCI-EXPANDED, SSCI, |
| Meganuclease | Title: (Meganuclease* or "Homing endonuclease*" or LAGLIDADG or I-SceI or I-CreI or I-DmoI) | | A&HCI |
| | | | CPCI-S |
| | | | CPCI-SSH |
| | | | BKCI-S |
| Transcription activator-like effector nucleases | Title: ("Transcription activator-like effector nuclease*" or TALENs or TALEN). | Timespan: All years | BKCI-SSH |
| | | | ESCI |
| Zinc finger nucleases | Title: ("Zinc finger nuclease*" or zfn*) | | |
| RNA interference | Title: ("RNA interference" or RNAi or "post-transcriptional gene silencing") | | |
| Viral vectors | ("*Viral vector*" or "adeno-associated virus" or lentivector*). | | |

6.  A structural network analysis was performed for each of the clusters were in order to obtain relevant metrics like the degree, indegree and outdegree of the papers.

7.  The content (title and abstract) of the top 20 percent of the papers with the highest degree (the number of received citations plus the number of the cited references) within each of the clusters was analyzed with KH Coder [28], a software for quantitative content analysis (text mining). KH Coder delivered a list of most distinctive terms for each cluster and a correspondence analysis.

8.  The most distinctive terms obtained from the text mining were searched in the manuscripts in order to provide context to these words. Then I used the words and context to formulate small phrases that could synthesize the general content of the clusters. Particularly, I sought to identify what could be the scientific problem around which the clusters were organized.

9.  The institutions and funding agencies related to the top ten percent of the papers with the highest indegree within each cluster were searched in the Web of Science. The research institutions were then classified into two categories: a) academic institutions or b) for profit private companies (Fundamentally pharmaceutical companies). The funding organizations were classified in two categories: a) government agencies and b) philanthropic organizations. Philanthropic organizations refers to organizations who are primarily grant-makers. For the purpose of this analysis, US based institutions who were required to file IRS form 990PF (required of all legal private foundations) were classified as philanthropic foundations. Private medical research institutions, hospitals and universities with 501(c)3 tax status (required to file IRS form 990) were classified as academic institutions.

10. For each of the time periods, Clust&see generated quotient graphs [27] in which clusters are represented as metanodes whose width is proportional to the number of their constituent papers. In this graphs the edges (Meta edge) represent the sum of the inter-citations between two clusters. Only the main interactions ($\geq 40$ inter-citations) between the clusters were shown. The information obtained from the text analysis and the information on the research institutions and founding agencies were integrated into the quotient graphs in order to build the maps of the structure and organization of the research for each of the time periods.

## Results

### The network model

A literature network of 16,746 papers and 138,168 inter-citations on viral vectors (Green nodes), RNAi (Blue nodes) and CRISPR/CAS (Red), meganucleases (Yellow), TALENS (Blush pink), and ZFNs (Brown) was constructed from the information downloaded from the Web of Science (Fig 1). The citation network consisted of 4113 papers on CRISPR/Cas, 304 on meganucleases, 288 on TALENs, 299 on ZFNs, 5071 on RNAi, 6401 on viral vectors and 270 papers reporting the combination of at least two of the studied technologies.

Two general features can be observed in the network model (Fig 1): The first one is the fact that papers tend to more frequently cite papers on the same technology than papers on different topics. The observed patron in the network model is related to homophilia which states that, based on node attributes, similar nodes may be more likely to attach to each other than dissimilar ones. That is, papers of the same technology appear clustered in the network model. The second relevant feature is that there are regions of contact among the investigation on these technologies- particularly a triple frontier among RNAi, viral vectors and genome editing research.

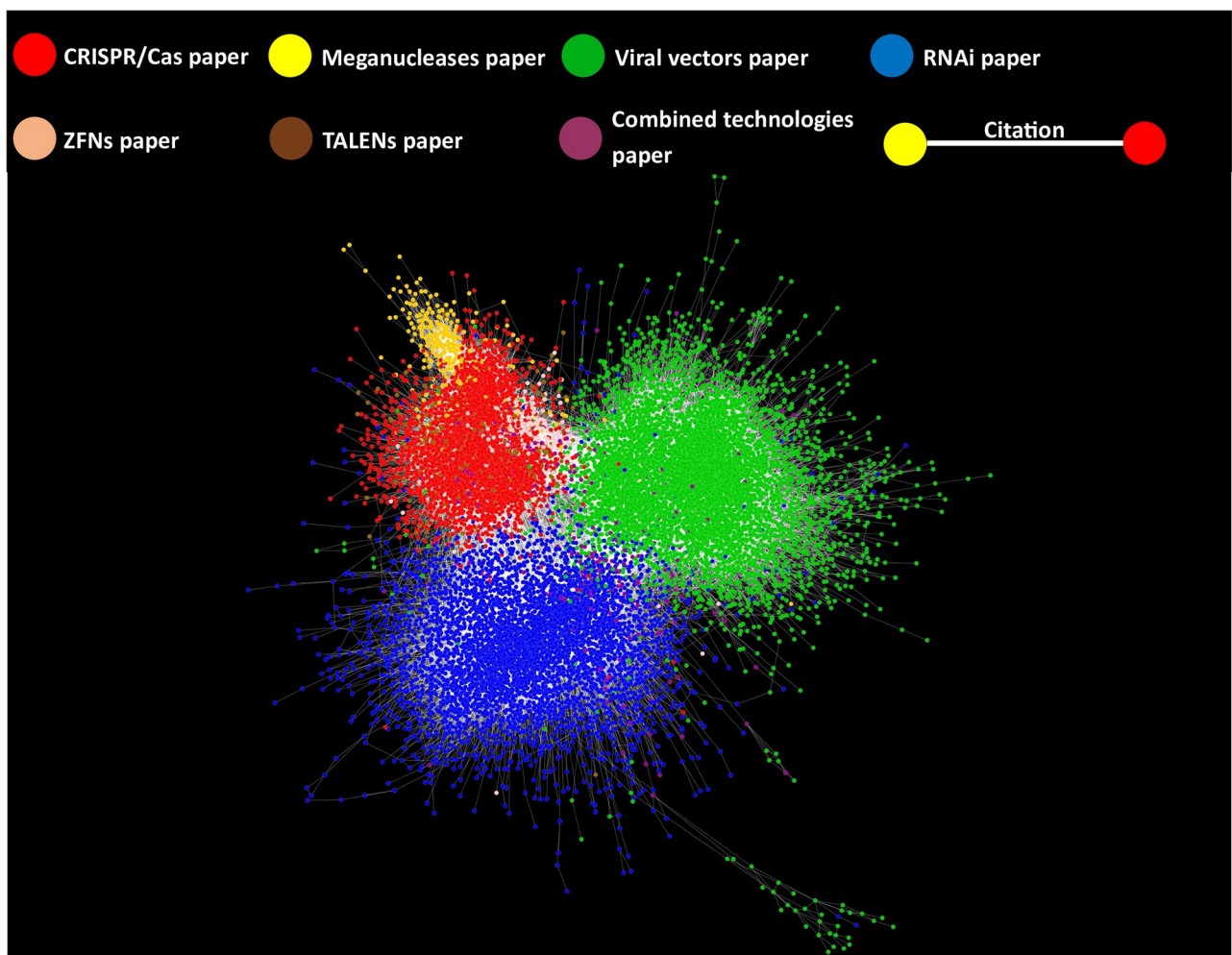

**Fig 1. Citation network model of papers on viral vectors (Green nodes), RNAi (Blue nodes) and CRISPR/CAS (Red), meganucleases (Yellow), TALENS (Blush pink), and ZFNs (Brown).** In this network model, nodes represent the papers on these genomic biotechnologies whereas edges (lines) represent the citations between these papers.

### The evolution of viral vectors, RNAi and genomic editing platforms

Fig 2 shows the evolution of research on RNAi and genomic editing platforms from 1980 to 2018 in terms of papers per year. It is important to notice that we have included in our search criteria terms that refer to biological entities and/or processes that predate the development of the studied technologies. Viral vector research is the oldest of the studied technologies. The investigation of this technology went through an exponential growth phase in the 1990s. From 2000 to 2006 viral vectors research exhibited a steady production of papers per year followed by a slow declining period from 2007 onwards. Similarly, RNAi research has exhibited stages of accelerated growth (2001–2005), stagnation (2006–2014) and decline (2014 onwards). On the other hand, CRISPR/Cas research has been in an exponential growth phase since 2012. In contrast to the above mentioned technologies, TALENs, homing endonucleses and ZFNs are much smaller research topics. The production of papers on these three genomic editing platforms reached its maximum between 2011 and 2014.

Figs 2 and 3 provide a mutually complementary scope of the coevolution of RNAi, viral vectors and genomic editing research. Nevertheless, a much more dramatic evolution of these

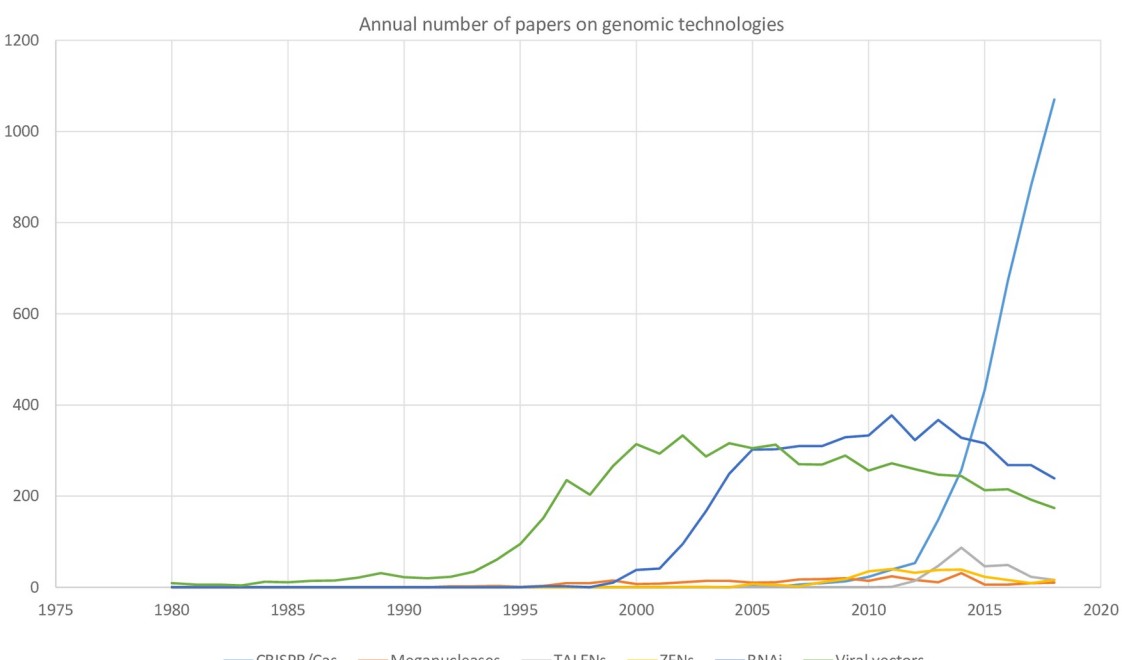

**Fig 2. Annual number of papers on viral vectors, RNAi and genomic editing platforms in the network model between 1980 and 2018.**

research topics can be observed in Fig 3. That is the succession of the genomic technologies is much more evident in terms of citation networks that in terms of the annual number of published papers. In the period between 1966 and 2001 (Fig 3), the giant component of the citation network is fundamentally made up of papers on viral vectors whereas research on RNAi and meganucleases are represented by two small regions in the network model. In the periods 2001–2005 (Fig 3) and 2006–2010 (Fig 3), the giant component is made up of mostly of papers on RNAi and viral vectors. Papers related to genomic editing platforms are in a peripheral position in the network model during these two periods (Fig 3). On the other hand, the periods 2011–2015 and 2016–2019 (Fig 3) are related to the consolidation and hegemony of genomic editing research respectively. Simultaneously, in the periods 2011–2015 and 2016–2019 the practical disappearance of RNAi research and viral vectors research can be observed. It is important to notice that the succession of genomic technologies is seen as more pronounced in the citation network model (Fig 3) because the papers that are not cited cannot be part of any network component.

A cluster analysis was performed for each one of the time periods in order to identify the community structure of the subnetworks. 53 network clusters (Newman modules) were identified. Each one of these clusters are related to specific stages in the evolution of these technologies: basic research, invention, improvement and application. A detailed description of these network clusters is provided in S1 Table. A correspondence analysis of the content (proper nouns, verbs and adjectives in the title and abstract of the papers) of these network clusters showing the general structure of the information in the network model was generated with KH Coder (S1 Fig). The correspondence analysis of the content showed that the network clusters are organized around the different technologies (viral vectors, RNAi and genome editing technologies) but also distinguished the research on different viral vectors (adeno-associated virus and lentivirus; S1 Fig). Interestingly, in the plot biomedical research terms are located at

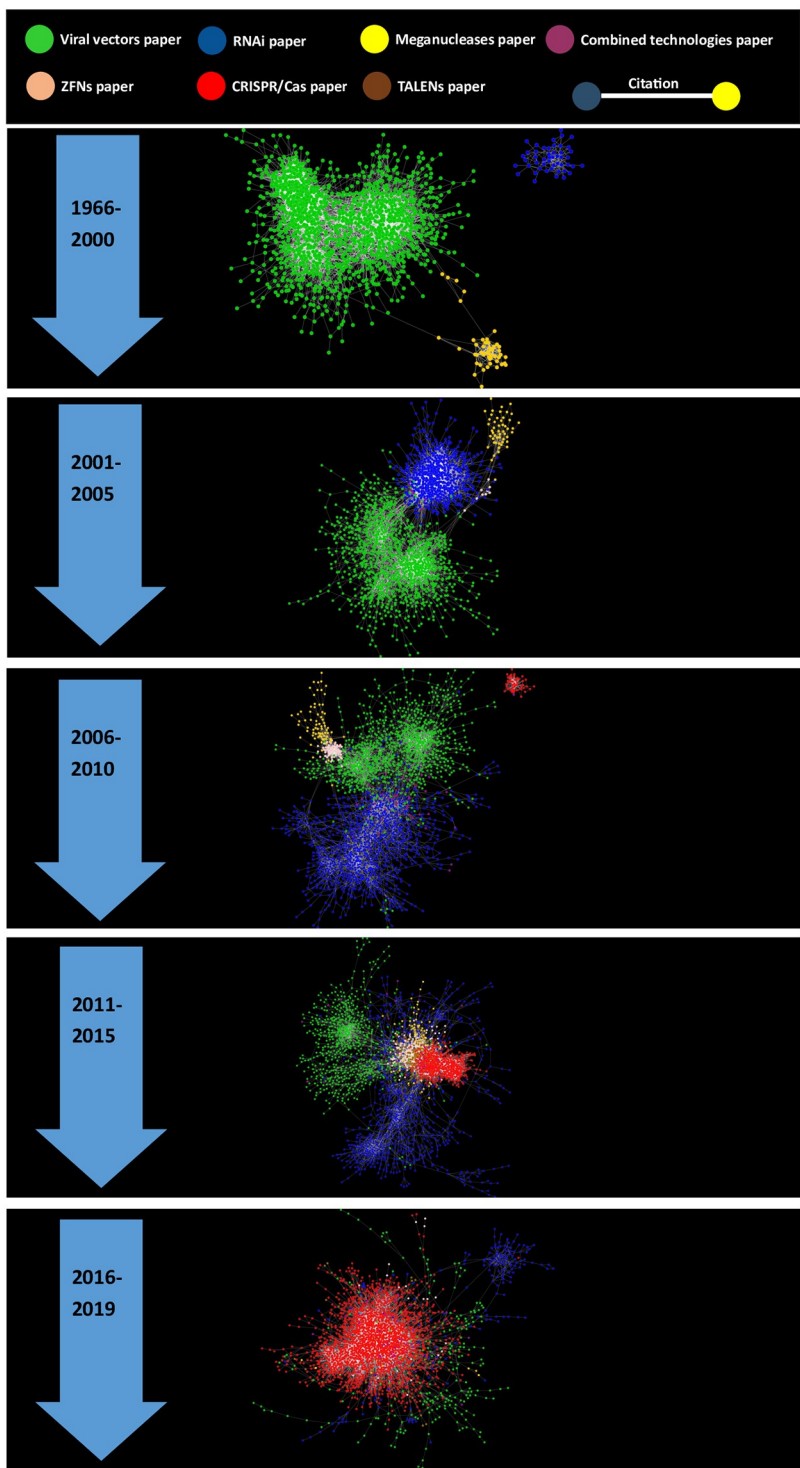

**Fig 3. Evolution of the citation network model of papers on viral vectors RNAi and genomic editing platforms.**
The network model shown in Fig 1 is divided into the following time periods: 1966–2000, 2001–2005, 2006–2010, 2011–2015 and 2016–2019. In each of the time periods only the giant component (the biggest subnetwork in which any two vertices are connected to each other by paths, and which is connected to no additional vertices in the rest of the network) is shown except for the periods 1966–2000 and 2006–2010 in which the second biggest component is shown, which is related to the emergence of RNAi and CRISPR/Cas respectively.

the center of the plot whereas terms related to the biomolecular bases of these technologies are located in the corners of the plot suggesting that these genomic technologies converge in their biomedical applications.

Five maps providing an overview of the organization, financing and leadership of the research on these technologies in each one of the time periods (1966–2000, 2001–2005. 2006–2010, 2011–2015 and 2016–2019) are described in detail below:

## Period 1966–2000

The first period (1966–2000) is entirely dominated by the investigation of viral vectors (Figs 2, 3 and 4). The network of clusters shows how viral vectors research is divided according to the type of vector, with adenoviral and adeno-associated vectors in one region in the network model, and retroviral (including lentiviral) vectors in the other. In this long period of time, the investigation of adeno-associated virus (AAV) vectors has transited from basic research on the molecular biology of the AVV virus in cluster 4–2000 to an invention phase related to the development of recombinant AVV vectors in cluster 2–2000 (Fig 4 and S1 Table). Following cluster 2–2000 in the citation network, a specialized community related to the invention and improvement of adenoviral vectors emerged in cluster 3–2000 (Fig 4 and S1 Table). This community is mainly focused on overcoming the limitations derived from the interaction of adenoviral vectors with the immune system. On the other region of the network model the papers are related to stages of invention in cluster 1–2000 and subsequent stages of improvement of retroviral vectors in clusters 5–2000 and cluster 6–2000 (Fig 4 and S1 Table). Two small clusters related to basic research on the mechanism of post-transcriptional gene silencing and the structure and function of homing endonucleases were also identified in this period (Fig 4 and S1 Table). The leading investigations (Top 10% of papers with the highest indegree) of each cluster were mostly performed in academic institutions with public funding in the United States (S1 Table).

## Period 2001–2005

In the period between 2001 to 2005, viral vectors research coexists and collaborates with RNA interference research (Figs 2, 3 and 4). In this period, RNAi research exhibits a logarithmic growth in terms of papers per year while viral vectors research is in a steady state (Fig 2). The network of clusters shows that in this period viral vectors research and RNA interference are connected through the interaction of cluster 3–2005 and cluster 4–2005 which are related to the development of these technologies in mammalian cells (Fig 4 and S1 Table). Actually, around 10 percent of papers in cluster 4–2005 ("RNA interference in mammalian cells") are either viral vectors research or the combination of these two technologies and thus a class of vectors that can be used as delivery systems for RNAi (S1 Table).

Viral vectors research in the period 2001–2005 is organized around different types of vectors (Lentiviral, AAV and adenoviral vectors) and the investigation of these technologies show different degrees of advancement (Fig 4 and S1 Table). While lentiviral vector research efforts are aimed to increase the efficiency of transduction in mammalian cells in cluster 3–2005, AAV vectors research is focused on the development of clinical applications (Gene therapy) in Cluster 1–2005, particularly in the identification of AVV serotypes which can differ in their tropism (Fig 4 and S1 Table). Meanwhile, adenoviral vectors research is, in this period, related to the development of helper-dependent adenoviral vectors for gene therapy (Fig 4 and S1 Table) which implied a significant improvement in terms of "large carrying capacity, reduced host adaptive immune responses and improved gene transfer efficiency" [29]. Finally, there is a small cluster of basic research papers mainly aimed to understand the mechanism of

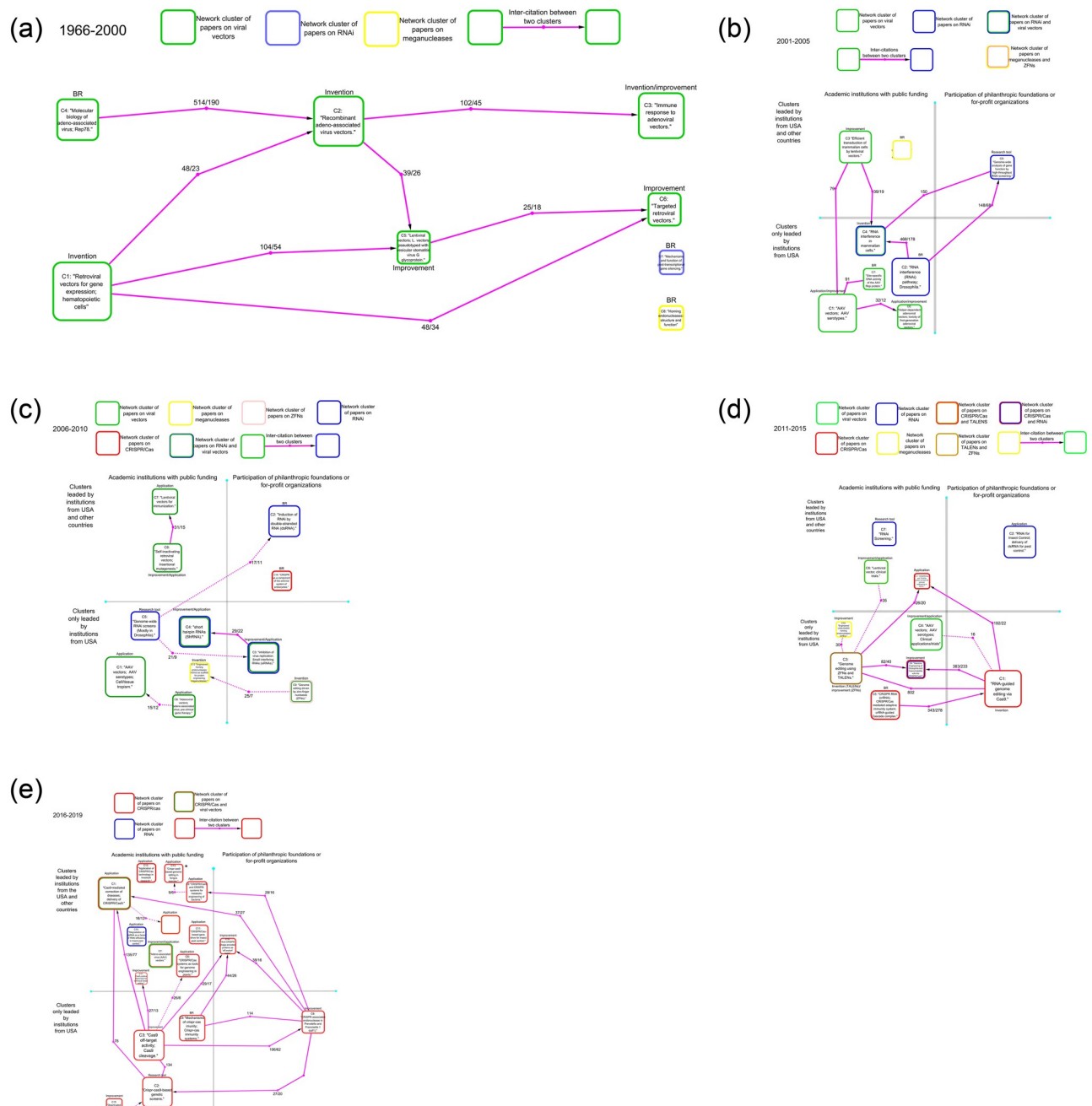

**Fig 4. Main interactions among the clusters of papers on viral vectors (green bordered rounded squares), RNAi (Blue) meganucleases (Yellow), TALENs (Blush pink), ZFNs (brown) and CRISPR/Cas (Red).** The networks of clusters in Fig 4 correspond to the following periods: 1966–2000 (A), 2001–2005 (B), 2006–2010 (C), 2011–2015 (D) and 2015–2018 (E). When a cluster is labelled with two colours it means it is composed by papers of two different types of genomic biotechnologies each one with at least 10% of the papers. The arrows represent the sum of the citations between two clusters showing the direction of the flow of information. The pair of numbers inside the arrows represent the number of times that each cluster cited another. If the number of inter-citations is equal in both directions, the arrow is replaced by an edge. The clusters are shown separately depending on whether the most cited investigations in the cluster were conducted only in publicly funded academic institutions or if they also included the participation of philanthropic foundations and/or for-profit organizations. Clusters were also separated according to whether the most cited investigations were conducted exclusively in US institutions or if these included the participation of organizations from other countries. When leading institutions or funding agencies could not be identified, the clusters were labelled with an asterisk. For a more detailed description of the clusters please see S1 Table.

integration of the AA virus that could lead to improve the long-term transgene expression of the AAV vectors (Fig 4 and S1 Table).

The clusters of papers on RNAi are organized around different stages of the investigation process (Fig 4 and S1 Table). Cluster 2–2005 is basic research aimed to identify and articulate the mechanism that conform the RNAi pathways (Fig 4 and S1 Table). Cluster 2–2005 represents a significant source of information for cluster 4–2005 which is related to the invention of RNAi as technology, and for cluster C5-2004 which use RNAi as a research tool that makes it possible to perform genome-wide analysis of gene function (Fig 4 and S1 Table). Cluster 10–2005 is a relatively small set of basic research papers related to the study of the structure and function of homing endonucleases Cluster 10–2005.

In the period 2001–2005 the leading research on viral vectors, RNAi and homing endonucleases was mostly performed by US academic institutions with public funding except clusters 3 ("Improvement of lentiviral vectors"), 5 ("RNAi as research tool") and 10 ("Basic research on homing endonucleases") which include European, UK and Canadian academic institutions. Financing for the leading research in cluster 5 includes the British philanthropic organization Wellcome Trust (Fig 4 and S1 Table).

Two general features can be observed in the network model (Fig 1): The first one is the fact that papers tend to cite more frequently papers on the same technology than papers on different topics. The observed patron in the network model is related to homophilia which states that, based on node attributes, similar nodes may be more likely to attach to each other than dissimilar ones. That is, papers of the same technology appear clustered in the network model. The second relevant feature is that there are regions of contact among the investigation on these technologies. Particularly a triple frontier among RNAi, viral vectors and genome editing research can be observed.

## Period 2006–2010

In the period between 2006 to 2010, viral vectors research entered a stage of decline while RNAi research stayed steady (Figs 2 and 3). In this period, the emergence of small clusters of papers related to CRISPR/Cas, homing endonucleases and ZFNs can be also observed (Fig 4 and S1 Table). In the period 2006–2010 the clusters of papers are poorly connected to each other compared with the previous periods (Fig 4). This could be related to a specialization process of viral vector and RNAi research in which the investigation is mostly oriented to the development of applications and specific improvement of these technologies.

The clusters of papers on viral vectors, which are organized around different types of vectors, appear practically disconnected from each other in this period (Fig 4 and S1 Table). Cluster 1–2010 and cluster 8–2010 are poorly connected to each other (Figs 3 and 4). These two clusters are related to the research on AAV and adenoviral vectors (S1 Table). Research in Cluster 1–2010, which is an extension of Cluster 1–2005, is focused on the preclinical and clinical utilization of different serotypes of AVV vectors in gene therapy (Fig 4 and S1 Table). Cluster 8–2010 is formed by papers on diverse aspects of the use of adenoviral vectors in gene therapy (S1 Table). The connection between cluster 6–2010 and 7–2010 is relatively stronger and there is a net flow of information form Cluster 6 to 7 (Figs 3 and 4). Research in cluster 6 is related to the use of a new generation of self-inactivating retroviral vectors aimed to reduce the genotoxicity observed in clinical studies. Papers in cluster 7 are mainly related to the use of lentiviral vectors for immunization.

In this period (2006–2010) four clusters of papers on RNAi were identified (Fig 4 and S1 Table). These four clusters are related to different stages of the investigation process: basic research, the use of RNAi as research tool, development of applications and the improvement

of RNAi technologies (Fig 4 and S1 Table). Cluster 2–2010 is basic research on the mechanism of RNAi induction (S1 Table). Cluster 3–2010 and cluster 4–2010 are both related to the diverse therapeutic uses of small interfering RNAs (siRNAs) and short hairpin RNAs (ShRNA) to inhibit gene expression (S1 Table). These two clusters include papers reporting the use of viral vectors mostly combined with siRNAs or ShRNA for gene therapy (S1 Table).

In this period (2006–2010) two clusters of papers related to the invention of homing endonucleases and ZFNs as gene editing technologies were identified (Fig 4 and S1 Table). Finally, a small cluster of basic research on CRISPR as a component of the antiviral system of prokaryote organisms was detected (Fig 4 and S1 Table).

Regarding the leading institutions in this period (2006–2010) it is important to notice that leading viral vectors research is performed in academic institutions with public funding (Fig 4 and S1 Table). The leading institutions for AVV and adenoviral vectors research are located in the United States whereas retroviral vectors and lentiviral vectors research include European academic institutions (Fig 4 and S1 Table). On the other hand, RNAi research includes the participation of one British philanthropic foundation (Wellcome trust) in cluster 2–2010 and one US private company (Alnylam Pharmaceuticals Inc) in cluster 3–2010 (S1 Table). In this period, ZFNs and CRISPR research is led by private companies in the USA (Sangamo BioSciences Inc) and the EU (Danisco France SAS) respectively (S1 Table). Leading research on homing endunucleases was completed in academic institutions with public funding (Fig 4 and S1 Table).

## Period 2011–2015

In the period between 2011 and 2015, viral vectors research continues its declining trajectory whereas RNAi research keeps steady in terms of papers per year (Figs 2 and 3). However, these two research topics (RNAi and viral vectors) are much smaller and less connected in terms of clusters of papers in the citation network (Figs 3 and 4). On the other hand, the investigation on the genomic editing platforms, particularly CRISPR/Cas9, entered a growing phase and the clusters of papers were strongly connected (Figs 2, 3 and 4).

Two clusters of papers on RNAi were identified in the period 2011–2015 (Fig 4 and S1 Table). The research reported in cluster 7–2015 specialized in the use of RNAi as a tool for the study of gene function whereas the use of RNAi as a tool in pest control was the main topic in cluster 2–2015 (Fig 4 and S1 Table).

Two clusters of papers on viral vectors were identified which are organized around the use of lentiviral (Cluster 6–2015) and AVV vectors (Cluster 4–2015) in gene therapy (Fig 4 and S1 Table). Both clusters include basic research papers, reports of technology improvement and preclinical and clinical studies (S1 Table).

In this period, the research on genomic editing research is organized in six clusters of papers (Fig 4): cluster 1–2015 is related to the invention of CRISPR/Cas as a genome editing platform (S1 Table). Cluster 5–2015 is basic research on the role of CRISPR RNA (crRNA) in the CRISPR/Cas mediated adaptive immunity systems (S1 Table). Cluster 9–2015 is related to the development of highly efficient methods to precisely engineer the Drosophila or Caenorhabditis genomes (S1 Table). This cluster includes also a significant number of papers on RNAi research in Caenorhabditis. Cluster 11–2015 is related to the application of CRISPR/Cas, TALENS and ZFNs platform for genome engineering of plant species (S1 Table). Cluster 3–2015 includes both research on the invention of TALENs as a genome editing tool and research aimed to improve ZFNs (S1 Table). Cluster10-2015 is related to the engineering of homing endonucleases as a genome editing tools (S1 Table).

The leadership in research and financing was held by academic institutions with private financing in the vast majority of the clusters during this period (Fig 4 and S1 Table). Only cluster 1–2015 "RNA-genome editing via Cas9"—which is related to the invention stage of this genome editing platform—and C2-2015 "RNAi for insect control"—which is a technological niche for RNAi—included the participation of philanthropic foundations (Fig 4 and S1 Table). The leading institutions of the most of the clusters were located in the USA (Fig 4 and S1 Table). Leadership in clusters 2, 6, and 7 include European Union research organizations and research in Cluster 11–2015 on applications of genome editing is led and financed by Chinese institutions (Fig 4 and S1 Table).

## Period 2016–2019

In the period between 2016 and 2019, the investigation on viral vectors and RNAi is represented by just two relatively small clusters of papers (Fig 4). In term of papers per year these two research topics are clearly declining (Fig 2). One of these network clusters is specialized in the use of RNAi technologies in pest control. Interestingly, a significant number of papers on viral vectors research report the use of this technology as a delivery system for CRISPR/Cas technologies (S1 Table). Similarly, research on the other genomic editing platforms (TALENs, ZFNs and meganucleases) has completely disappeared in terms of organized clusters of papers (Fig 4). The remaining investigation on these platforms has been absorbed by CRISPR/Cas research (Fig 4 and S1 Table).

CRISPR/Cas research continues in an exponential growth stage in terms of number of papers per year (Fig 2). In this period CRISPR/Cas research is highly complex and diverse (Fig 4 and S1 Table). Seven clusters of papers organized around different areas of application were identified: five clusters organized around specific technical improvements, a basic research cluster and one cluster on the use of CRISPR/Cas as research tool in functional genomics (Fig 4 and S1 Table). The interaction among the clusters provides a relevant picture of the current organization of CRISPR/Cas research as follows:

Clusters 3–2019 and 5–2019 are net suppliers of information for the rest of the clusters (Fig 4). Cluster 5–2019 is basic research related to the study of the mechanisms of CRISPR/Cas immunity and the discovery and analysis of the different CRISPR/Cas immunity systems, whereas Cluster 5–2019 is formed by a set of investigations mostly aimed to improve the specificity and fidelity of Cas9 nucleases (Fig 4 and S1 Table). These two clusters supply information to clusters related to technical improvements (Clusters 8–2019, 16–2019 and 17–2019) and clusters related to specific areas of application (Clusters 1–2019 and 6–2019; Fig 4 and S1 Table). Following these two clusters in the information flow is Cluster 8–2019 which is related to the discovery and development of the CRISPR-associated endonuclease in Prevotella and Francisella 1 "Cpf1" (Fig 4 and S1 Table). Cpf1 is a technology analogous to the CRISPR/Cas9 system [30]. Cluster 8–2019 received information from Cluster 3–2019 and supplied information to clusters 1–2019, 2–1019 and 9–2019 which are related to specific areas of application of CRISPR technologies (Fig 4 and S1 Table). At the end of the information flow in the network model there are a set of clusters related to the following areas of application and improvement: Cas9-mediated correction of diseases, CRISPR/Cas9-based genetic screens, CRISPR/Cas systems as tools for genome engineering in plants, CRISPR/Cas9 and CRISPRi systems for metabolic engineering of bacteria, CRISPR/Cas9 based genome editing in fungus species, anti-CRISPR phage-encoded proteins as 'off-switch' tools and Cas9-cytidine deaminase fusion for individual bases editing (Fig 4 and S1 Table). Two CRISPR/Cas Clusters are practically disconnected to one another: Cluster 11–2019, which is related to the use of CRISPR/Cas-based gene drives for insect pest control and Cluster 12–2019 which is related to the application of

CRISPR/Cas technology in livestock research (Fig 4 and S1 Table). Finally, Cluster 15, which is related to the investigation of activation/deactivation of CRISPR/Cas9, supplied information to Cluster 2–2019 (Fig 4 and S1 Table).

In this period the leading research in each cluster is largely performed in academic institutions with public financing (Fig 4 and S1 Table). Only two clusters include the participation of philanthropic foundations financing leading research: Cluster 8–2019, related to the discovery of Cpf1 which is a technology analogous to the CRISPR/Cas9 system, and Cluster 16–2019 whose research is mostly aimed to use of bacteriophage proteins to modulate CRISPR/Cas (Fig 4 and S1 Table). Interestingly, most of the clusters related to areas of application include leading institutions outside the US, mainly in China and the European Union, whereas the basic research on the CRISPR/Cas mechanism, the use of CRISPR/Cas as a research tool and the efforts aimed at the technological improvement of CRISPR technologies are led by US institutions (Fig 4 and S1 Table). That is, in this period, an important part of the strategic knowledge on CRISPR/Cas9 produced in publicly funded US academic institutions was used by academic institutions that were at least partially funded by philanthropic organizations in order to achieve much more specific improvements (Fig 4 and S1 Table). Subsequently, academic institutions in China, Europe and the US used the basic knowledge and the technologies developed in the US in order to apply these technologies in biomedicine, plant engineering, metabolic engineering, pest control and livestock (Fig 4 and S1 Table).

## The interaction zone among the genomic technologies

An interaction zone of 4,706 papers and 11,078 cross-citations connecting the genomic technologies was identified in the citation network model. This region is formed by papers reporting the combined use of these technologies or papers exhibiting cross-citations among the genomic technologies. Fig 5 describes the evolution of the interaction zone in terms of the number of papers and citations received. The evolution of the number of papers on each technology in the interaction zone resembles the evolution of the whole network model when Fig 5 is compared with Fig 2. However, the annual number of TALENs and ZFNs papers are of a comparable size to RNAi and viral vectors in the recent years in the interaction zone, suggesting an increasing interaction among the genome editing platforms. The annual number of papers reporting the combined use of genomic technologies exhibits a steady growth but it seems that it is not boosted by the exponential growth of the number of papers on CRISPR/Cas technologies (Fig 2). Plot B of Fig 5 provides a clearer view of the impact of earlier genome editing platforms (TALENs and ZFNs) in the subsequent development of CRISPR/Cas technologies as the peaks of the citations received by ZFNs and TALENs papers published in 2010 and 2013 respectively were of a size only comparable to the number of CRISPR/Cas papers in the interaction zone. The matrix of interactions in S2 Table shows the information flows between the genomic technologies via cross-citations. As expected, our analysis showed that CRISPR technologies are a net "consumer" of the knowledge produced by the investigation on the other genomic technologies (Fig 5 and S2 Table). The research fields on ZFNs and TALENs were by far the main source of information for CRISPR/Cas research (Fig 5 and S2 Table) followed by the research on RNAi and viral vectors (Fig 5 and S2 Table). The interaction between viral vectors research and RNAi research was the third strongest interaction having its maximum between 2006 and 2008 (Fig 5 and S2 Table). From a broader perspective, our results point to a decrease in the interaction between the genomic technologies as the older ones are displaced by the CRISPR/Cas revolution.

(a)

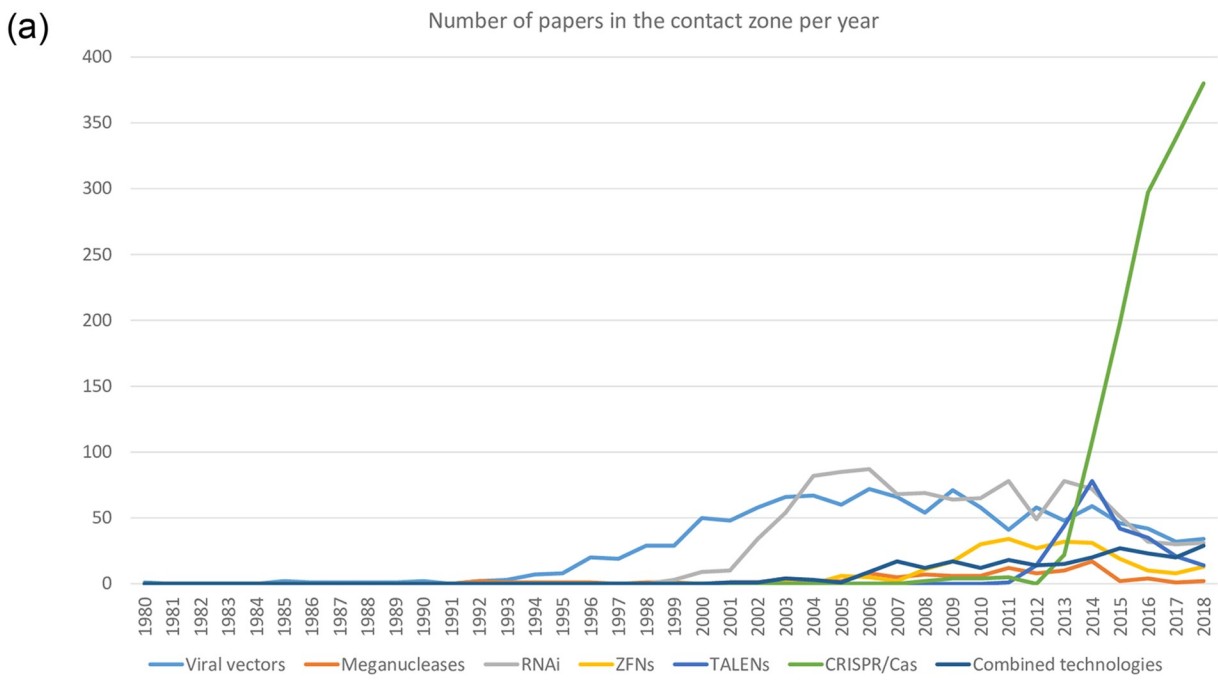

(b)

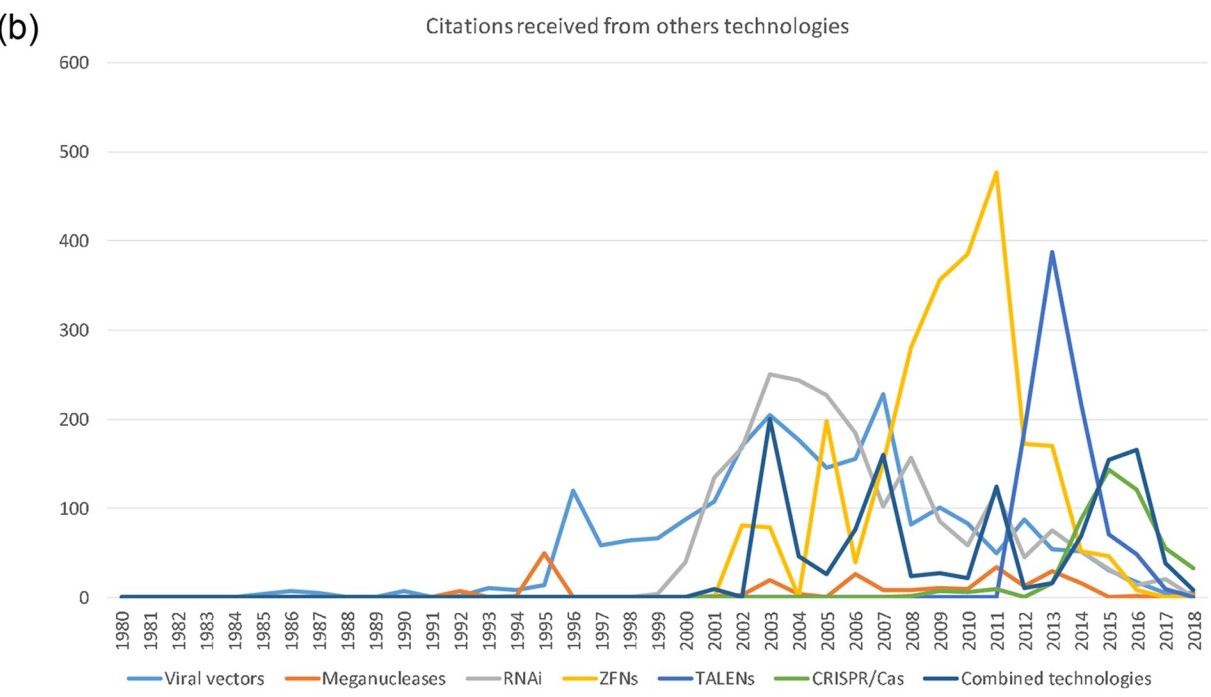

**Fig 5. Evolution of the interaction zone between 1980 and 2018.** A) Number of papers in the contact zone per year. B) Citations received from other technologies.

## Discussion

The identified clusters of papers relate to different stages in the investigation process: basic research, invention, use of the technologies as research tools, improvement, and applications. That is, the identified clusters relate to specialized communities of inventors, improvers, and potential innovators. By tracking the emergence and the flow of information among the clusters of papers, different trajectories from basic research to applications can be tracked. These trajectories provide a general scope of the coevolution of the research on viral vectors, RNAi and genome editing platforms. Our results broadly agree with recent reports in the literature in terms of the main challenges, milestones and areas of application [31–34]. Even the observed collaboration between viral vectors and other technologies has been reported in detail in various reviews [13, 14]. The obtained maps offer a complementary perspective to those of review papers on the historical evolution of the research on these technologies. However, the most important contribution of the present investigation is not to provide an already well documented historical view on these technologies. Instead, the maps are an excellent tool to visualize the organization of research on these genomic biotechnologies as a simultaneously scientific and socio-political process.

In that sense, it is important to consider that papers with the highest indegree inside each cluster of papers have a key role as organizers of these regions in the literature network [35]. That is these papers, which receive the most citations from the cluster they belong to, define the technical problems to solve and the horizons of possible applications and future investigations [35]. Therefore, the research institutions and funding agencies with more papers in the top ten most cited papers within each cluster have an outsized impact on the invention, improvement and application of these technologies.

In this investigation, the strategy of grouping the clusters according to the type of funding (exclusively academic institutions with public funding vs participation of philanthropic foundations or for-profit organizations) and the location (only USA entities vs participation of institutions from other countries) of the investigations with the greatest impact within each of the specialized research communities allows us 1) to visualize how the global subsystem of investigation on biotechnologies has evolved, and 2) to formulate a hypothesis on the historical and geopolitical factors influencing the investigation of the studied biotechnologies.

The main observation that can be made when reviewing the generated maps is that the investigations on viral vectors, RNAi and CRISPR/Cas exhibit different trajectories in terms of 1) their translational process (from basic research to specific areas of application), 2) the geopolitical distribution of leading institutions, and 3) the patterns of research funding. Before explaining what these differences consist of, it is necessary to consider that these observations are made exclusively from the perspective of scientific literature networks and therefore exclude the information contained in databases on clinical trials, patents and approved products, all which could lead to a more comprehensive vision about the evolution of these technologies. However, recent analyses of patents related to genomic editing technologies complement our results [36, 37]. The differences between these technologies in terms of the trajectory of the translational process, the geopolitical distribution of leading institutions and the patterns of research are described as follows:

1. Our results showed that the clusters of papers on viral vectors are organized around the two main types of vectors: those based on adeno-associated virus and those based on lentivirus. Therefore, there are two independent "translational" trajectories with a relatively week interaction with each other. Following the invention phase in viral vector research, there is a long term process of incremental innovation in which the rest of the clusters are organized around specific improvements of viral vectors as essential components of gene

therapy. The comparison of our results with the published reviews [31, 38] on viral vectors lead us to think that the results of preclinical and clinical studies are the leading force of this process of incremental innovation in viral vectors research. Currently, the clinical applications of viral vectors exhibit a promising degree of success, Voretigene neparvovec, a AAV vector-based gene therapy was approved by the FDA in 2018 [39], and LentiGlobin, a lentiviral vector gene therapy, was approved by the European Medicines Agency in 2019 [40]. Additionally, it is important to note that viral vectors have shown a high grade of flexibility as it could be incorporated in delivery systems for RNAi and CRISPR technologies [13, 14]. Regarding location and research financing, the leading research on viral vectors has been fundamentally sustained with public funding through the different stages of the translational trajectory while the leading institutions were located entirely in the United States and in the European Union.

2. The "translational" trajectory of RNAi research results appears to be different from that of viral vectors since RNAi research included, throughout its evolution, clusters of papers related to the study of RNAi as a biological phenomenon and clusters related to the use of RNAi as a research tool for the study of gene function. Interestingly, one of the two clusters detected in the period 2016–2019 is related to the use of RNAi for pest control, whereas in the same period no cluster of papers organized around a biomedical application was identified, which implied that RNAi research has reached a specialized application niche due to the strong competition that CRISPR could represent in the biomedical field. Even though the implementation of RNAi as a therapeutic technology has faced serious difficulties, an RNAi-based therapy was recently approved by the FDA in 2018 [32]. The latter could strengthen the development of new RNAi-based therapies. Focusing on the location and financing of leading research on RNAi, the participation of Wellcome Trust financing basic research and Alnylam Pharmaceuticals Inc. as the sole leading organization found in cluster 3–2010 (applied research), suggests a transition from a model of research performed in academic institutions with public financing to a mixed model in which philanthropy and private companies play a strategic role in the development and application of these technologies.

3. The trajectory of genome editing technologies, particularly CRISPR/Cas9, is much more complex and diversified than viral vectors and RNAi research. This trajectory begins modestly with the emergence of small clusters of basic research followed by clusters related to the invention phase of each of the genomic editing platforms. However in the two last periods (between 2011 and 2019), a process of diversification and articulation of CRISPR/Cas9 research can be observed. In this process, clusters related to basic research, improvements and a diversity of areas of application coexist and articulate. Basic research leads to the development of technical improvements, and these two in turn provide the information required for the development of biomedical, agricultural, livestock and industrial biotechnology applications. In that sense, the present results agree with a patent analysis [37] and reviews on CRISPR/Cas9 [8, 33, 34] to suggest that the success of this technology is the product of the long term accumulation of a diversity of previous technological developments. A geopolitical division was observed in which leading institutions in basic research and technical improvements were located in the United States and the development of applications took place mainly at or in collaboration with Chinese institutions. The participation of leading institutions in the European Union was relatively small in CRISPR/Cas research. Recently, a "geographical bias" in the global CRISPR patent landscape was reported, which is in line with our results. That is, the leadership in terms of number of patents related to technical improvements remained in the USA whereas a tight competition in the area of biomedical applications between China and USA was reported [36].

According to this patent analysis, China has dramatically overtaken all other nations in patents for CRISPR/Cas9 in agricultural, livestock and industrial applications [36]. Our analysis further suggests a concentration of philanthropic organizations financing the invention phase and certain strategic improvements in CRISPR/Cas9 research. Additionally, our results suggest that several of the most influential investigations in the invention stage of ZFNs were conducted by a pharmaceutical company: Sangamo BioSciences Inc. However, leading basic research and some improvements were financed exclusively with public resources in the USA, whereas the leading research in the different areas of application was financed by Chinese government agencies.

These differences reflect the historical changes in the world economic and political system that have taken place in the last forty years and are characterized by 1) The consolidation of the "neoliberal" economic model in the financing of science [41] and 2) the emergence of China as a scientific superpower that rivals the United States and the European Union [42]. The fact that the leading basic research for all three genomic technologies was entirely publicly funded by US government agencies broadly agrees with Mazzucato's analysis of the 'entrepreneurial state' where breakthrough technologies emerge from long term state investments in public missions [43]. Conversely, the financing of RNAi and CRISPR/Cas9 applications conforms to a "neoliberal" model of financing in which the state funds basic research, and the financing for applied research is left to philanthropic foundations and private companies [44, 45]. This model is the dominant force in the United States, where about 55% of overall R&D funding for applied research now comes from the private sector, while 34% comes from government [46]. 85% of the money available for R&D in the US is from the private sector [46]. Anne-Emanuelle Birn documented in detail the transition toward this neoliberal model of science financing in the field of public health in the 90s and 2000s. Birn showed how the increased power of philanthropic foundations and pharmaceutical companies have modified the global health research agenda as these entities are now occupying the place that previously belonged to the state, altering the priorities and trajectories of global health research [41]. Our results suggest something similar may be occurring in CRISPR/Cas research in the United States as the applications for publicly funded basic research are increasingly determined with the participation of private sector and philanthropic actors.

The recently published report "Research and Development: US Trends and International Comparisons" which is part of the Science and Engineering Indicators 2020 of the National Science Board provides more information about the general context of research and development (R&D) in the United States in which genomic technologies, particularly CRISPR/Cas is been developed and applied [49]. The annual US R&D expenditure has increased from $406,579,000 in 2010 to $547,886,000 in 2017. This growth has been largely due to the business, non-profit organization and higher education sectors as the participation of the US federal government in the financing of R&D has remained stable. However, when US R&D expenditure is measured as a percentage of the gross domestic product, there is a clear relative decline of federal participation [49]. Additionally, the report analyses research and development expenditures in 2017 by sector, source of funds, and type of work. Most of the 2017 expenditures go to experimental developments in the business sector ($347,622,000). This quantity is almost 9 times larger than federal investment in basic research ($38,653,000) [49].

Publicly funded institutions in China lead in the application of genomic editing technologies. Regarding the emergence of China as a main player in CRISPR/cas9 research it is important to mention this state investment is nothing new; almost a decade prior to the emergence of these technologies China was considered the top leader in DNA sequencing. [47] Furthermore, in 2016 China launched a $9 billion effort to boost the development of precision

medicine—around 25 times the budget that Barack Obama proposed to devote to the development of the same technology [48]. In a broader perspective, it has been suggested that China, in its path to achieve global technological leadership, has targeted biotechnology as a "strategic emerging technology" which implies "large-scale government grants, tax concessions, easy access to bank loans, and supportive policies regarding intellectual property, standardization, and human resources" aimed to promote the industrial development of these technologies [42].

Regarding the interaction between the genomic technologies our results suggest that even though the previous technologies, especially ZFNs and TALENS, were highly relevant to the development of CRISPR/Cas research, the current trend is toward a minor level of interaction between the studied technologies as the development and diversification of CRISPR/Cas technologies is displacing the older technologies. In a previous investigation, by comparing the knowledge translation process of two different cancer nanotechnologies, liposomes versus metal nanoparticles, we showed the enormous difficulty of combining two or more radical technological innovations [15, 16]. Liposomal anti-cancer technologies showed a higher degree of success in their translation process (from basic research to clinical applications) than metal nanoparticles [15]. Both nanotechnologies are considered potential radical innovations. However, liposomes can be used as delivery system of small molecules drugs while the use of metal nanoparticles in anti-cancer therapies is bases in the physical destruction of the cancerous tissues [15, 16]. That is, even though liposomes in anti-cancer therapies could be considered a radical innovation, these technology could move from a stage of invention to a stage of innovation because it was possible to articulate this technology with the more traditional pharmaceutical approach [15, 16]. Interestingly, we found several clusters of papers related to alternative uses of liposomes to treat cancer like combining with biomolecules and other nanoparticles [15, 16]. However, none of these alternative proposal succeeded, as far as we know, in being translated into approved treatments. Once Doxil (doxorubicin liposomal pegylated) success as the first liposomal therapy approved by the US Food and Drug Administration (FDA), an incremental innovation process aimed to improve and amplify the therapeutic performance of doxorubicin liposomal pegylated followed the FDA approval [16]. Something similar could be happening in the case of genomic technologies. Because of the success of CRISPR is linked to its ease of use and versatility compared with former genomic technologies the combined use of CRISPR with other genomic technologies would add extra complexity and make it more difficult to successfully deliver technologies. However, given that the research fields on ZFNs and TALENs were by far the main source of information for CRISPR/Cas (followed by the research on RNAi and viral vectors), it reiterates again the importance of long term basic research funding, the vast majority of which in this case was provided by US government agencies.

Finally, it is important to mention that a deeper understanding of the processes of articulation of scientific research around these potentially revolutionary technologies is of strategic interest for society. That is, different trajectories in the evolution of these technologies can lead to different technological outcomes which in turn could meet the needs of different countries, companies or population groups. The total dominance of US public funding for basic research and the relative absence of democratically controlled institutions in leading research on CRISPR/Cas applications will affect what applications are developed, and for whose benefit.

## Conclusion

Our results suggest that the three studied technologies exhibit differences in the global articulation of their translational trajectories (from basic research to applications). In each of the different stages of the evolution of viral vector research, the most influential investigations were

conducted in academic institutions in the United States and the European Union whereas the main funders of the top cited research within each cluster were government agencies. In the case of RNAi research, a philanthropic organization and a pharmaceutical company played a key role in the development of basic RNAi research and clinical application respectively, in addition to government agencies and academic institutions. Finally, the trajectory of CRISPR/ Cas9 research, the most complex and diverse, exhibits a geopolitical division of the investigation efforts between the US, as the main producer of the more cited basic research and technical improvements, and China increasingly leading the applied research. Even though USA and Chinese government agencies funded most of the top cited research within several clusters, several philanthropic foundations played an important role in the financing of research in the stages of invention and technical improvements.

Mapping citation networks and major funding agencies allows us to visualize innovation as a socio-political process. The different observed trajectories in the development of genomic editing technologies could reflect the historical changes in the world economic and political system that have taken place in the last forty years and are characterized by 1) the consolidation of the "neoliberal" economic model in the financing of science, where the state funds limited basic research and leaves applications to the private sector and 2) the emergence of the Chinese state as a scientific superpower. These results raise further questions about research translation, innovation policy and the role of public institutions in shaping the applications of publicly funded innovations.

## Supporting information

**S1 Table. Description of clusters of papers.**
(PDF)

**S2 Table. Matrix of interactions (cross-citations) among the genomic technologies.** The parentheses in the matrix indicate the average year of publication of the documents in each interaction. The interactions with more of 1,000 cross-citations are highlighted in red while interactions with more of 500 cross-citations are highlighted in pink.
(XLSX)

**S1 Fig. Correspondence analysis plot of the content of the network clusters.**
(PDF)

## Author Contributions

**Conceptualization:** David Fajardo-Ortiz, Annie Shattuck, Stefan Hornbostel.

**Data curation:** David Fajardo-Ortiz.

**Formal analysis:** David Fajardo-Ortiz.

**Funding acquisition:** David Fajardo-Ortiz, Stefan Hornbostel.

**Investigation:** David Fajardo-Ortiz, Annie Shattuck.

**Methodology:** David Fajardo-Ortiz.

**Project administration:** David Fajardo-Ortiz, Stefan Hornbostel.

**Resources:** David Fajardo-Ortiz.

**Supervision:** Stefan Hornbostel.

**Visualization:** David Fajardo-Ortiz.

**Writing – original draft:** David Fajardo-Ortiz.

**Writing – review & editing:** Annie Shattuck.

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
