## [Decision Letter · Decision Letter 0]

4 Feb 2020

PONE-D-19-35217

Mapping the coevolution, leadership and financing of research on viral vectors, RNAi, CRSPR/Cas9 and other genomic editing technologies

PLOS ONE

Dear Dr. Fajardo-Ortiz,

Thank you for submitting your manuscript to PLOS ONE. After careful consideration, we feel that it has merit but does not fully meet PLOS ONE’s publication criteria as it currently stands. Therefore, we invite you to submit a revised version of the manuscript that addresses the points raised during the review process.

We would appreciate receiving your revised manuscript by Mar 20 2020 11:59PM. To enhance the reproducibility of your results, we recommend that if applicable you deposit your laboratory protocols in protocols.io, where a protocol can be assigned its own identifier (DOI) such that it can be cited independently in the future. For instructions see: http://journals.plos.org/plosone/s/submission-guidelines#loc-laboratory-protocols

We look forward to receiving your revised manuscript.

Kind regards,

Omaththage P. Perera, Ph.D.

Academic Editor

PLOS ONE

Journal Requirements:

Reviewers' comments:

Reviewer's Responses to Questions

**Comments to the Author**

1. Is the manuscript technically sound, and do the data support the conclusions?

Reviewer #1: Yes

Reviewer #2: Yes

2. Has the statistical analysis been performed appropriately and rigorously? 

Reviewer #1: Yes

Reviewer #2: Yes

3. Have the authors made all data underlying the findings in their manuscript fully available?

Reviewer #1: No

Reviewer #2: Yes

4. Is the manuscript presented in an intelligible fashion and written in standard English?

Reviewer #1: No

Reviewer #2: Yes

5. Review Comments to the Author

Reviewer #1: I applaud the efforts of Ortiz et al. to put together an interesting comparative piece evaluating 6 different genome editing/interference technologies. The paper touches on a relevant and impactful topic, and the authors have made a good effort at presenting a coherent story. While the analysis performed is overall sound and interesting, I have a few comments that should help improve the piece further:

1. My main concern is with the way the figures are presented. Figures 1, 3 and even to an extent 4, are hard to go through. The network plots especially don't convey much new information that wouldn't be intuitive or better summarized through a few lines of text. I would strongly urge the authors to restructure their figures to make them easy to understand and follow.

2. Often networks are not robust to perturbations in minor features of the observations. For instance, one question the authors can ask here is how similar are the citation records to each other based on which of the 6 topics is the primary theme of the paper. Recently, an interesting paper in Science proposed a method to do this: https://advances.sciencemag.org/content/5/12/eaau9630/tab-article-info (Semblance, Science Advances, Dec 2019)

I encourage the authors to try this in the current work as I believe it can enhance their results and the insights they draw.

3. When the authors are talking about an evolution of the different methods as it relates to the 1990s and early 2000s, a discussion from other fields can help make their argument stronger. For instance, in physics or chemistry, are there similar trends where a technology was speculated to be obsolete for several decades, but a new technology helped revive the old technology instead of completely overshadowing it?

4. The network model that the authors have built is interesting, but on its own it conveys little new insights. Similar to my comment #2 above, the authors should address whether the model can be used for any predictive tasks, such as predicting citation records, or inferring the number of nodes one might have to traverse to reach from theme A (for eg., CRISPR) to theme B (for example, RNAi).

5. Lastly, the authors have done a good job of building a chronological timeline from early 20th century to the present day. I think similar to my comment #1 above, a figure or table containing salient events from each decade would be a good graphic that the authors should consider including.

Overall, I found the article write interesting and it will be relevant to a broad range of scientists. Once the authors address the concerns above, this paper has my support for publication in PLoS One.

Reviewer #2: The manuscript presents an observational assessment of the growth of three different biological systems for altering organismal genetics during the past forty years. The authors mined information from 16,746 manuscripts examining those that employed viral vectors, RNAi and gene editing (including ZFN’s, TALENs and CRISPR/CAS systems). The interactive parameters examined were the citation cross-referencing along with the geopolitical and institutional funding of the research. The data presented shows the reported activities establishing and utilizing each of these biotechnologies, thus providing an indication of the volume of research reported for each biotechnology and its application. The data are presented in structural graphs that establish the biotechnology cluster and the cross-referencing matrix. In general, the manuscript is well written and the is topic covered well but there are omissions of analysis that would improve the manuscript.

There are some components of the information that should be expanded. The manuscript has not included the recently published National Science Foundation National Science Board NSB-2020-1 report which has additional information on the funding sources and global research activities and capabilities that are discussed in this manuscript. The manuscript should address the findings from NSB report as well. The data should also provide the authors with information on the patents and applications of these biotechnologies. For example, the use of RNAi has been of limited use in medicine but in agriculture has expanded for control of various pests. The authors do not address this component of the application of this biotechnology. The application of CRISPR/CAS systems to agriculture is expanding and it again would be useful to have this information segregated and highlighted from the data presented. In addition, there is an expanding application of gene editing in various insects to provide pharmaceutical materials that is occurring in foreign markets that was not discussed that could be included.

6. PLOS authors have the option to publish the peer review history of their article (what does this mean?). If published, this will include your full peer review and any attached files.

Reviewer #1: No

Reviewer #2: Yes: Paul D Shirk

---

## [Author Response · Author response to Decision Letter 0]

22 Mar 2020

Omaththage P. Perera, Ph.D.

Academic Editor

PLOS ONE

Dear Dr. Perera,

Attached please find the revised version of our manuscript entitled “Mapping the coevolution, leadership and financing of research on viral vectors, RNAi, CRISPR/Cas9 and other genomic editing technologies. First of all, the authors would like to thank you for a very professional and careful peer review, which has certainly added to our manuscript.

We have carefully addressed the Reviewer´s observations as follows:

Reviewer #1 

“I applaud the efforts of Ortiz et al. to put together an interesting comparative piece evaluating 6 different genome editing/interference technologies. The paper touches on a relevant and impactful topic, and the authors have made a good effort at presenting a coherent story. While the analysis performed is overall sound and interesting, I have a few comments that should help improve the piece further:”

Re: The authors appreciate this positive comment. Indeed, our intention with this manuscript was to provide a clear perspective on the evolution of these genomic technologies through the processing of a considerable amount of scientific information.

“My main concern is with the way the figures are presented. Figures 1, 3 and even to an extent 4, are hard to go through. The network plots especially don't convey much new information that wouldn't be intuitive or better summarized through a few lines of text. I would strongly urge the authors to restructure their figures to make them easy to understand and follow.”

Re: We improved the quality of figures 1, 3 and 4 by adding legends explaining the meaning of the elements of the graphics. Also we added some arrows pointing the time direction in figure 3. This helped immensely. 

“Often networks are not robust to perturbations in minor features of the observations. For instance, one question the authors can ask here is how similar are the citation records to each other based on which of the 6 topics is the primary theme of the paper. Recently, an interesting paper in Science proposed a method to do this: https://advances.sciencemag.org/content/5/12/eaau9630/tab-article-info (Semblance, Science Advances, Dec 2019) I encourage the authors to try this in the current work as I believe it can enhance their results and the insights they draw.”

Re: We added an analysis of how similar the content of the network clusters are through a correspondence analysis of the content (proper nouns, verbs and adjectives in the title and abstract of the papers) of these network clusters showing the general structure of the information in the network model. The correspondence analysis of the content showed that the network clusters are organized around the different technologies (viral vectors, RNAi and genome editing technologies) but also distinguished the research on different viral vectors (adeno-associated virus and lentivirus). Interestingly, in the plot, biomedical research terms are located at the center of the plot whereas terms related to the biomolecular bases of these technologies are located in the corners of the plot suggesting that these genomic technologies converge in their biomedical applications. Use of the kernel “semblance” for the analysis of the content of the papers would be a good addition, but is beyond our technical capabilities and to some degree the paper’s scope. We hope the addition of the data on similarity will address this concern. 

“When the authors are talking about an evolution of the different methods as it relates to the 1990s and early 2000s, a discussion from other fields can help make their argument stronger. For instance, in physics or chemistry, are there similar trends where a technology was speculated to be obsolete for several decades, but a new technology helped revive the old technology instead of completely overshadowing it?”

Re: We briefly included in the discussion the case of the innovation process of two anti-cancer nanotechnologies: liposomes versus metal nanoparticles. These instances are particularly interesting as both nanotechnologies are considered radical innovations. Liposomes can be used as delivery system of small molecules drugs while the use of metal nanoparticles in anti-cancer therapies is bases in the physical destruction of the cancerous tissues. That is, even though liposomes in anti-cancer therapies could be considered a radical innovation, liposome-based therapies were more successful in progress in its innovation process than metal nanoparticles because it was possible to articulate this emerging technology with the more traditional pharmaceutical approach. 

“The network model that the authors have built is interesting, but on its own it conveys little new insights. Similar to my comment #2 above, the authors should address whether the model can be used for any predictive tasks, such as predicting citation records, or inferring the number of nodes one might have to traverse to reach from theme A (for eg., CRISPR) to theme B (for example, RNAi).”

Re: We addressed this point by analysing the evolution of a very interesting region in the citation network model. This region is made of 4,706 papers and 11,078 cross-citations which connect the genomic technologies. The papers in this region either report the combined use of genomic technologies or papers exhibit cross-citations among these technologies. The analysis of the evolution of the flows of information across this region pointed out that even though the previous technologies, especially ZFNs and TALENs, were highly relevant to the development of CRISPR/Cas research, the current trend is toward a minor level of interaction between the studied technologies as the development and diversification of CRISPR/Cas technologies is displacing the older technologies. 

“Lastly, the authors have done a good job of building a chronological timeline from early 20th century to the present day. I think similar to my comment #1 above, a figure or table containing salient events from each decade would be a good graphic that the authors should consider including.”

Re: In the introduction and discussion sections we cited several reviews that provide information about the milestones in the history of each of the studied technologies. The paper already has too many figures, so we are hesitant to add more when that information has been covered in other reviews. 

Reviewer #2

“The manuscript presents an observational assessment of the growth of three different biological systems for altering organismal genetics during the past forty years. The authors mined information from 16,746 manuscripts examining those that employed viral vectors, RNAi and gene editing (including ZFN’s, TALENs and CRISPR/CAS systems). The interactive parameters examined were the citation cross-referencing along with the geopolitical and institutional funding of the research. The data presented shows the reported activities establishing and utilizing each of these biotechnologies, thus providing an indication of the volume of research reported for each biotechnology and its application. The data are presented in structural graphs that establish the biotechnology cluster and the cross-referencing matrix. In general, the manuscript is well written and the topic is covered well but there are omissions of analysis that would improve the manuscript.”

Re: We appreciated this positive comment as it confirms that the paper was clearly understood by reviewers. 

“There are some components of the information that should be expanded.The manuscript has not included the recently published National Science Foundation National Science Board NSB-2020-1 report which has additional information on the funding sources and global research activities and capabilities that are discussed in this manuscript. The manuscript should address the findings from NSB report as well. The data should also provide the authors with information on the patents and applications of these biotechnologies. For example, the use of RNAi has been of limited use in medicine but in agriculture has expanded for control of various pests. The authors do not address this component of the application of this biotechnology. The application of CRISPR/CAS systems to agriculture is expanding and it again would be useful to have this information segregated and highlighted from the data presented. In addition, there is an expanding application of gene editing in various insects to provide pharmaceutical materials that is occurring in foreign markets that was not discussed that could be included.”

Re: We discussed the results of the recently published report “Research and Development: US Trends and International Comparisons” which is part of the Science and Engineering Indicators 2020 of the National Science Board. This report provides key information about the general context of research and development (R&D) in the United States in which genomic technologies, particularly CRISPR/Cas has been developed and applied. This was a very useful addition. Thank you!

We hope this version satisfy the concerns of the reviewers. Thank you again for such a collegial and productive review. The paper is much better for your efforts! 

Sincerely,

David Fajardo-Ortiz

---

## [Editor Report · Decision Letter 1]

25 Mar 2020

Mapping the coevolution, leadership and financing of research on viral vectors, RNAi, CRISPR/Cas9 and other genomic editing technologies

PONE-D-19-35217R1

Dear Dr. Fajardo-Ortiz,

We are pleased to inform you that your manuscript has been judged scientifically suitable for publication and will be formally accepted for publication once it complies with all outstanding technical requirements.

With kind regards,

Omaththage P. Perera, Ph.D.

Academic Editor

PLOS ONE

---

## [Editor Report · Acceptance letter]

27 Mar 2020

PONE-D-19-35217R1 

Mapping the coevolution, leadership and financing of research on viral vectors, RNAi, CRISPR/Cas9 and other genomic editing technologies 

Dear Dr. Fajardo-Ortiz:

I am pleased to inform you that your manuscript has been deemed suitable for publication in PLOS ONE. Congratulations! Your manuscript is now with our production department. 

With kind regards,

on behalf of

Dr. Omaththage P. Perera 

Academic Editor

PLOS ONE